# Prevalence and Predictors of Anxiety among Stable Hospitalized COVID-19 Patients in Malaysia

**DOI:** 10.3390/ijerph20010586

**Published:** 2022-12-29

**Authors:** Muhammad Azri Adam Bin Adnan, Mohd Shaiful Azlan Bin Kassim, Norhafizah Bt Sahril, Mohamad Aznuddin Bin Abd Razak

**Affiliations:** Institute for Public Health, National Institutes of Health, Ministry of Health Malaysia, Shah Alam 40170, Malaysia

**Keywords:** COVID-19, anxiety, mental health, hospitalised, Malaysia

## Abstract

The COVID-19 pandemic has created anxiety among hospitalized SARS-CoV-2 patients. Therefore, this study aimed to determine the prevalence of anxiety and its associated factors among stable inpatient COVID-19 patients in Malaysia. **Method:** A cross-sectional study was conducted using a web-based online survey involving 401 patients from Malaysia’s leading COVID-19 hospitals from 15th April until 30th June 2020, who were chosen using quota sampling. The General Anxiety Disorders 7 items (GAD-7) scale, the Coping Orientation to Problems Experienced Inventory (Brief-COPE) and a socio-demographic profile questionnaire were used. Descriptive analysis and multiple logistic regression were performed using SPSS v23 to determine the prevalence of anxiety and its associated factors. **Result:** The results showed that the prevalence of anxiety was 7.0%. Multiple logistic regression analysis revealed that female gender (*p* < 0.05), a fear of infection (*p* < 0.05), a lack of information (*p* < 0.05), a maladaptive coping mechanism of behavioral disengagement (*p* < 0.001) and self-blame (*p* < 0.001) were significantly associated with anxiety. Meanwhile, adaptive coping mechanisms via instrumental support (*p* < 0.001) were a significant protective predictor of anxiety. **Conclusions**: COVID-19 infection has had a significant influence on the mental health of patients. Findings in our study provide baseline data on the prevalence of anxiety among stabilized COVID-19 inpatients in Malaysia. Despite the relatively low prevalence, the data have the potential to improve the present mental health monitoring system and the deployment of suitable treatments in dealing with similar circumstances.

## 1. Introduction

An outbreak driven by a novel coronavirus has become the focus of scientific attention in recent months. Severe acute respiratory syndrome coronavirus 2 (SARS-CoV-2) is a strain of coronavirus that causes the transmission of coronavirus disease 2019 (COVID-19). In March 2020, the World Health Organization (WHO) proclaimed the outbreak as a pandemic and declared that the world was in the midst of a worldwide health crisis [1], as the disease is strongly linked with very serious health issues and can be deadly [2]. Beginning in November 2019, this pandemic, which began in Wuhan, China, spread to 216 nations and, to date, more than 25 million people have contracted the disease, and it has caused more than 5 million deaths worldwide [3]. On 24 January 2020, Malaysia announced its first case of COVID-19. Since then, there have been over 2 million positive cases discovered [4].

The pandemic has been linked to multiple social disturbances as well as severe economic effects. These factors, together with the possibility of stigma and prejudice, can contribute to mental health issues for patients who contract COVID-19, as well as for the public [5,6,7]. The World Health Organization (WHO) has recommended that mental health issues should be taken seriously and monitored during the extended COVID-19 response as one of its essential health services [8]. Mental well-being assessments conducted in China following COVID-19, as early as March 2020, reported a significant increase in negative emotions such as despair, anxiety, and dread of death in both the affected and unaffected populations. The population as a whole was assessed to have a decreased level of enjoyment and a strong sense of insecurity. This anxiety became more severe among COVID-19 patients, resulting in self-isolation, feelings of despair, and fear of infecting others, even after they had been treated and had completed the quarantine period [9]. During the third wave of this pandemic, on 20th September 2020, a cross-sectional online survey was conducted involving 1544 participants, using social media platforms to study the prevalence and determinants of depression and anxiety in the Malaysian population. The results showed that 43.6% of the participants had symptoms of anxiety, with 34.1% of them categorized as a mild to moderate anxiety level. In term of depression, it was discovered that nearly three quarters of the participants were depressed, with 70% of the respondents suffering from moderate to severe depression [10].

COVID-19-related mental health issues in the general population, healthcare workers, and those with a diagnosed mental disorder were the topics of previous research investigations. The study on the mental health impacts of COVID-19 on affected patients is still limited, owing to the fact that, in infection units, the patient’s physical well-being has traditionally taken precedence over their psychological assessment. Patients with COVID-19 may be under more psychological stress than the general population because of their treatment in isolation wards, and they may experience boredom and loneliness during the quarantine period. Additionally, the circumstances that they encounter while hospitalized can be traumatizing [11]. Nervousness and anxiety are also frequently seen in isolation and quarantine wards [12]. In a recent comprehensive study by Brooks et al. 2020, it was observed that unfavorable psychological consequences such as post-traumatic stress symptoms, bewilderment, and rage are some of the psychological effects of quarantine. Longer isolation periods, infection concerns, frustration, and boredom were but a few of the stressors that impacted the patients [12]. As proven during the outbreaks of influenza A (H1N1), Ebola virus disease (EVD), severe acute respiratory syndrome (SARS), and Middle East respiratory syndrome (MERS), hospitalized patients experience significant psychological distress both during the acute illness and in the long-term phase of the disease after an epidemic [13,14,15,16]. 

In a study conducted by A. Zandifar et al. in Iran, a high prevalence of psychiatric disorders among hospitalized COVID-19 patients was discovered. It was reported that 100% of the patients who participated in the study experienced anxiety. Furthermore, it was highlighted that 97.2% of COVID-19 patients had some level of depression, while 97.1% of them had some degree of stress [17]. Additionally, in one of the earliest studies on the mental health effects of COVID-19, researchers at a hospital in Wuhan, China, found that the prevalence of anxiety and depression symptoms among hospitalized patients with COVID-19 was 18.6% and 13.4%, respectively [18]. 

COVID-19 hospitalized patients’ mental health is a major issue. This is a global problem deserving worldwide attention. Hence, the purpose of this study was to determine the prevalence of anxiety among inpatient COVID-19 patients through their demographic characteristics, which could provide a further understanding and awareness of the importance of addressing mental health in this group. The American Psychological Association (APA) defines anxiety as “an emotion characterized by feelings of tension, worried thoughts, and physical changes” [19]. Anxiety, which is known to be a strong predictor for suicidal ideation and attempts [20], together with its associated risk factors among hospitalized COVID-19 patients, must be assessed in order to provide early psychiatric interventions during the hospitalization period. Therefore, this study also planned to develop a predictive model for anxiety among hospitalized COVID-19 patients and their associated predictors.

## 2. Methods and Materials

### 2.1. Study Design

This was a cross-sectional study conducted in selected COVID-19 hospitals in Malaysia from 15 April 2020 until 30 June 2020. The selected hospitals were: (i) Hospital Kuala Lumpur (HKL), (ii) Hospital Permai Johor Bahru (HPJB), (iii) Hospital Sungai Buloh (HSB), and (iv) the Malaysia Agro Exposition Park Serdang low-risk patient quarantine and treatment center (MAEPS). The target population consisted of all COVID-19 patients who were admitted to the hospitals. The sampling framework comprised all individuals diagnosed with COVID-19 infections who were admitted to general wards and were in a stable condition. The quota sampling technique was used in this study, with the first 428 eligible patients responding to the screening being recruited. Screening of eligible responses commenced on the 15 of April 2020. A respondent had to be at least 18 years old, be diagnosed with COVID-19, have stable health (non-intensive care unit), and have been admitted for more than 24 h in a general ward or quarantine center. Additionally, they had to be fluent in Malay or English and able to converse in both languages. Participation in this study was entirely optional, as the respondents could refuse or withdraw at any time throughout the survey. If a respondent withdrew, their whole profile, including any responses, was removed.

### 2.2. Sample Size and Study Procedure

A total of 401 COVID-19 patients participated in this study. The data collection was conducted using an online platform—Google Forms—to prevent the transmission of COVID-19. The respondents were contacted by research assistants, who provided them with information about the study, and we afterwards acquired their agreement via the Google Forms questionnaire in the first section. Approximately 15 to 20 min were required to complete the self-administered online questionnaire, which was available in both the English and Malay languages. A pilot study with at least 30 different responses was performed to validate the system, and changes were made based on the feedback from the survey takers. The completed methods of this study and its technical report can be found in [21].

### 2.3. Ethics Approval and Privacy

This study was registered under the National Medical Research Registry (NMRR), Ministry of Health Malaysia (NMRR-20-711-54541), and it obtained ethical approval from the Medical Research and Ethics Committee (MREC), Ministry of Health Malaysia.

The database was protected to ensure the privacy and confidentiality of the data. By assigning a unique password to each file, the dataset was protected in any format. Only researchers from the core team were permitted to examine the participants’ personal information, and the data will be preserved.

### 2.4. Survey Instruments/Questionnaire

For the data collection of this research, a structured questionnaire in Malay and English was built into a Google Forms online survey. The survey included a sociodemographic part as well as specific study instruments. The Google Form was divided into four sections, with the first one being the patient information sheet and consent page. The second section of the Google Form included information on the respondents’ sociodemographic characteristics and stressors associated with mental health. Age group, gender, marital status, education level, occupation, ethnicity, and family income were all included in the sociodemographic data. In terms of stressors, the fear of infection, social discrimination, financial burden, and lack of information about COVID-19 were assessed in this study. The third section included GAD-7 questions that were used to screen possible anxiety. The fourth section was the coping strategies using the Brief Coping Orientation to Problems Experienced (Brief-COPE) scale. The two psychometric instruments (GAD-7 and Brief-COPE) used in the questionnaires, both in English and Malay, had previously been validated locally [22,23].

### 2.5. Generalized Anxiety Disorder Scale (GAD-7)

The GAD-7 is a 7-item self-report questionnaire that is often used in primary care and mental health settings to screen for the presence of anxiety. It assesses the presence of anxiety symptoms in the preceding two weeks of everyday living. Each item had four answers: (i) not at all, (ii) several days, (iii) more than half the days, and (iv) nearly every day. Each of the seven items was scored from 0 (not at all) to 3 (nearly every day). Scores of the GAD-7 ranged from 0 to 21. Total scores of 8 and above indicated the existence of anxiety [24]. Spitzer et al. [25] invented the original version of the tool. Sherina et al. later validated it in the Malay form [23]. The Malay version of GAD-7 had a sensitivity of 76% (95% CI 61–87%), a specificity of 94% (88–97%), positive LR 13.7 (6.2–30.5), and negative LR 0.25 (0.14–0.45), and it was found to have good internal reliability (Cronbach’s alpha = 0.74) [23].

### 2.6. Brief Coping Orientation to Problems Experienced (Brief-COPE) Inventory

Brief-COPE is a 28-item self-report questionnaire derived from the original 60-item COPE Inventory. It is used to evaluate coping mechanisms in reaction to stress [26]. In total, 14 dimensions are covered by this scale. These are self-distraction, active coping, denial, substance use, use of emotional support, use of instrumental support, behavioral disengagement, venting, positive reframing, planning, humor, acceptance, religion, and self-blame. Every dimension has two items rated on a four-point Likert scale, ranging from “I haven’t been doing this at all” (score of 0) to “I have been doing this a lot” (score of 3). Each of the 14 subscales represents a distinct pattern of coping, which may be either adaptive or maladaptive. The adaptive coping subscale contains 16 items with a possible range of 0 to 48, such that higher scores indicate greater use of adaptive coping. The adaptive coping subscale includes active coping, planning, positive reframing, acceptance, humor, religion, using emotional support, and using instrumental support. The maladaptive coping subscale contains 12 items with a possible range of 0 to 36, such that higher scores indicate greater use of maladaptive coping. The maladaptive coping subscale includes self-distraction, denial, venting, substance use, behavioral disengagement, and self-blame. In Malaysia, both the English and Malay versions have been validated [27]. The validated Brief-COPE’s internal consistency indicated by the Cronbach’s alpha values ranged from 0.25 to 1.00, while the intraclass correlation coefficient (ICC) ranged from 0.05 to 1.00 [27].

### 2.7. Definition of Variables

Major ethnic groups in Malaysia include the Malay, Chinese, and Indian populations, with indigenous communities and local people of Sabah and Sarawak falling under the “other Bumiputras” category. Those who did not have Malaysian citizenship were classified as “others”, which included legal and illegal immigrants.

The level of education was determined using the Malaysian educational system. The highest level of education attained in a public or private institution that offered formal education was referred to as the education level. Those who had never attended school in any of the educational institutions that provided formal education were considered as having no formal education. Primary education refers to Standard 1 through 6 or equivalent. Secondary education was defined as having completed Form 1 through 5 or its equivalent, and tertiary education was defined as having earned a diploma or other higher credentials.

B40, M40, and T20 refer to the household income classification in Malaysia. B40 represents the bottom 40%, M40 represents the middle 40%, and T20 represents the top 20% of Malaysian household income. The B40, M40, and T20 household incomes are below RM4850 per month, between RM4851 and RM10,970 per month, and over RM10,971 per month, respectively.

### 2.8. Data Analysis

The Statistical Package for Social Sciences (SPSS) Version 23.0 for Windows was used to analyze the data. Descriptive statistics were used to assess the prevalence of anxiety and demographic characteristics using a chi-square test. Multiple logistic regression was used to construct a prediction model for anxiety. All variables with *p*-value < 0.25 in univariable analysis and variables known to be associated with anxiety from published articles were included in the model to control possible confounding factors. The backward LR step was used to obtain the best predictor model for anxiety. A *p*-value < 0.05 was considered to be significantly associated with anxiety.

## 3. Results

### 3.1. Socio-Demographic Characteristics

A total of 401 COVID-19 patients from three hospitals (Hospital Sungai Buloh, Hospital Permai JB, Hospital Kuala Lumpur) and MAEPS were involved in this study. Around 68% of the participants were from Hospital Sungai Buloh, 15% from MAEPS, 11% from Hospital Permai JB, and 6% from Hospital Kuala Lumpur. The demographic information of the COVID-19 patients (respondents) is shown in Table 1. The majority of the respondents were males aged 18–34 years old. The mean age (SD) of the participants was 32.65 (11.58) years old. In terms of ethnicity, the Malays made up the majority of the respondents (68.3%), followed by others (18.2%), Chinese (6.7%), Indians (5.0%), and other Bumiputera (1.7%). Approximately 49.1% of the participants were married, with 50.9% of them being single or widowed. In terms of education, 50.4% had completed tertiary education, 25.4% had completed secondary school, 16.1% had completed elementary school, and 8.2% had not received a formal education. Most of the respondents worked in the private sector (34.4%), followed by being self-employed (12.5%), public servants (8.5%), and healthcare professionals (8.5%). Others were either unemployed, retired, or students (37.7%), or housewives (2.0%). The majority of the respondents (71.8%) were in the B40 household income group, followed by M40 (17.0%) and T20 (11.2%). In addition, 81.8% of the participants in this survey were Malaysians, while 18.2% were non-Malaysians.

### 3.2. Prevalence of Anxiety by Hospital

Overall, the prevalence of anxiety among hospitalized COVID-19 patients was 7.0% (*n* = 28), with the mean GAD-7 score being 2.58 (SD, 3.58). The highest anxiety prevalence rate was recorded by Hospital Permai JB (9.1%), followed by Hospital Sungai Buloh (8.1%), MAEPS (3.2%), and Hospital Kuala Lumpur (0%).

### 3.3. Prevalence of Anxiety by Socio-Demographic Characteristics

Table 2 summarizes the prevalence of anxiety and its associations with socio-demographic characteristics. Anxiety was significantly more likely in the single/widow/er group (*p* < 0.05) than in married participants. Male respondents were shown to have a greater prevalence of anxiety than female respondents, and respondents from the age group of 18–34 had the highest prevalence rate when compared with other age groups. In terms of ethnicity, the Chinese had the highest prevalence rate. Additionally, anxiety was shown to be higher among Malaysian nationals than among non-Malaysian nationals. Anxiety was not significantly prevalent in any specific education level group or among occupation groups. Furthermore, in this study, there was no significant relationship between anxiety and household income group.

### 3.4. Prediction Models for Anxiety

Subsequently, simple and multivariable logistic regression analyses were used to assess the risk factors for anxiety in connection to socio-demographic variables, stressors, and coping methods in COVID-19 patients. Bivariable analysis found that anxiety was associated with those patients who were single/widow/ers, those not working/students/pensioners, and those in the B40 household income category. In term of stressors, the fear of infection and lack of information were found to be significantly associated with anxiety. Self-distraction, active coping, denial, substance use, behavioral disengagement, venting, humor, and self-blame are coping strategies that are associated with anxiety. Further multivariable analysis using logistic regression revealed that respondents who were single/widowed had 2.87 times higher odds of having anxiety than those who were married (aOR = 2.87, 95% CI: 1.01, 8.18). Furthermore, in terms of stressors, the fear of infection (OR = 1.82, 95% CI: 1.08, 3.04) and a lack of information (aOR = 1.82, 95% CI: 1.08, 3.04) were both found to be major risk factors associated with anxiety. In addition, instrumental aid as a coping method was significantly correlated with a decrease in anxiety (aOR = 0.65, 95% CI: 0.47, 0.90). However, dysfunctional coping mechanisms such as behavioral disengagement (aOR = 2.03, 95% CI: 1.30, 3.18) and self-blame (aOR = 1.74, 95% CI: 1.31, 2.30) were found to be associated with anxiety in this study. Refer to Table 3 below.

## 4. Discussion

Initially, little attention was paid to the influence of the COVID-19 pandemic on mental health, but a large number of research articles on this topic have already been published. Nadir S. et al. reported a 31.9% prevalence of anxiety in 17 studies with a total sample size of 63,439 in a recent systematic review and meta-analysis article [28]. These figures are significantly higher than the pre-pandemic era, when the prevalence of anxiety was estimated to be 3.6% according to a WHO report [29]. However, despite the expanding volume of scientific research on mental health during the COVID-19 pandemic, there are still substantial information gaps. According to earlier research, COVID-19 patients are more likely than the general public or healthcare workers (HCW) to have negative mental health symptoms [30,31,32]. However, research on mental health during the COVID-19 has rarely included COVID-19 patients.

The current study is based on a technical report on research conducted in four local hospitals in Malaysia to evaluate the mental health status of stable COVID-19 patients who are hospitalized [21]. The technical report revealed that the prevalence of depression, anxiety, and suicidal ideation was 7.5% (*n* = 30), 7.0% (*n* = 28), and 4.0% (*n* = 16), respectively. In addition, it was shown that fear of infection was one of the most significant contributors to mental health, with the largest number of “agree” and “strongly agree” responses (52%), followed by discrimination (43.9%), financial burden (40.9%), and a lack of information (33.1%). Multiple logistic regression was also employed, in addition to the descriptive analysis, to develop a prediction model for anxiety and its associated risk factors in hospitalized COVID-19 patients.

In this cross-sectional study, we found that the prevalence of anxiety among hospitalized patients with COVID-19 in Malaysia was 7.0%, which was lower when compared to the prevalence globally [33,34,35]. Furthermore, the prevalence of anxiety reported in Malaysia also appeared to be lower when compared with other Asian countries, such as India (17.49%) [36], South Korea (18.0%) [37], and Bangladesh (30.7%) [38]. Our study also revealed a lower prevalence of anxiety among hospitalized COVID-19 patients compared to the prevalence reported in China, which was 16.4% [39]. The initial pandemic that occurred in China in November 2019 subsequently registered a higher number of cases in the country compared to Malaysia. This severe outcome is most likely the reason that China has a far greater incidence of anxiety compared to Malaysia.

Our results revealed a substantially higher prevalence of anxiety than the 1.7% national prevalence reported in the National Health and Morbidity Survey (NHMS) in 2011 among the general population [40]. This demonstrates that the COVID-19 pandemic has had a significant effect on the mental health of patients. However, the influence of the COVID-19 pandemic on mental health has varied across time, with mental health symptoms being much more severe towards the onset of the pandemic and significantly less severe in the months that followed [41], indicating that a degree of resilience in mental health may be emerging in reaction to the pandemic [42].

This prediction model study reported that marital status is the main predictor of anxiety. Our results found that patients who are single/widowed are three times more likely to develop anxiety compared to those who are married. Our findings corroborate research done in Spain [43] and China [44] among COVID-19 populations, which reported that a divorced or widowed status was associated with poor mental health and anxiety. As expected, living alone without a partner is associated with an elevated risk of loneliness, which is exacerbated in situations of social and physical isolation. Furthermore, a survey of older adults in London found that being widowed or divorced increased the likelihood of experiencing worsening components of anxiety after the COVID-19 lockdown [45].

During the COVID-19 pandemic, the public were exposed to an unusual environment of danger and uncertainty. Individuals who were isolated physically and socially, had a fear of infection, and were exposed to other stressors may have been particularly susceptible to anxiety-related symptoms [12,46,47]. According to our study, the primary stressors contributing to anxiety are the fear of COVID-19 infection, and the lack of information on COVID-19. Fear is one of the most important causes in the development of emotional issues such as anxiety and stress. This is supported by a previous Indian study that discovered that a fear of COVID-19 infection was the most common cause of suicide cases in India during the COVID-19 outbreak [48]. The availability of the internet in spreading news rapidly, especially fake and false information, may contribute to the rise in fear and consequently anxiety levels [49]. In fact, it has been shown that anxiety can be reduced by having access to the most up-to-date and correct information sources [50]. In this regard, mental health practitioners advise using only official information sources and ignoring information obtained from unauthorized channels and unregulated sources [51].

Our study revealed that maladaptive coping strategies such as behavioral disengagement and self-blame were significantly associated with anxiety. These findings are consistent with those of prior research, which found that under the COVID-19 lockdown restrictions, maladaptive coping methods such as behavioral disengagement and substance abuse raised anxiety levels [52]. Self-blame typically arises after a stressful incident that has the potential to lead to negative results, and in which one attributes responsibility, causation, and/or intentionality to oneself. Self-blame is frequently the result of a mistaken cognition [53]. On the other hand, instrumental support was strongly linked with anxiety reduction in our study. Adaptive coping may have aided the creation of stress buffers, improved psychological well-being, and improved overall health outcomes [54]. These results are in line with those of recent research, which found that adaptive, instrumental, and social coping techniques were linked to improved stress management and reduced negative mental health consequences [55].

### Strengths and Limitations

This study has several strengths and limitations. To the best of our knowledge, this is the first research work performed to examine anxiety levels among hospitalized COVID-19 patients in Malaysia. This research offers early information on the mental health status of COVID-19 patients who were hospitalized in Malaysia during the pandemic, which should draw the interest of policymakers, health facility administrators, and any individuals engaged in the response to COVID-19 or any future epidemic. Besides the above, the findings of the present study add evidence to the current literature specifically on the effects of mental health in inpatient COVID-19 individuals.

The limitations included the research being performed using a cross-sectional design, which meant that the cause-and-effect relationships between the numerous factors in the study could not be established. COVID-19’s mental health effects should be studied further to determine their long-term implications. There is a pressing need for additional extensive investigations, such as cohort studies or interventions, to be conducted in the future. Second, due to the wide spread of the disease, it is recommended that research with a larger sample size should be conducted in more hospitals to obtain more accurate data. Thirdly, although this was a multi-center study, the recruited patients were only from the main hospitals located in town areas. Thus, in terms of national representation, these data are not representative of all patients with COVID-19 in Malaysia.

## 5. Recommendations

The findings of this study indicate that integrated mental health interventions are required for patients infected with COVID-19. Firstly, health officials must identify high-risk populations based on socio-demographic information. Based on this study, single/widowed individuals are the target population with the highest risk of anxiety. Information on early coping strategies and social support must be provided for them, as this will alleviate their anxiety. Secondly, public health interventions can be improved by establishing a mental health surveillance system via online platforms or telemedicine. Furthermore, health authorities should encourage the public and patients themselves to seek COVID-19 information only from trusted sources. Combating false COVID-19 information and disseminating accurate scientific information will help patients to better understand the virus, reducing their fear and the negative perception of COVID-19. Lastly, providing a supportive hospitalization environment and improving communication between patients and doctors, especially regarding the clinical progression and prognosis of the disease, will maintain a relatively stable, healthy level of psychological and physical functioning among patients.

## 6. Conclusions

COVID-19 infections have negative impacts on patients’ mental health. The current study provides some valuable data on the prevalence of anxiety among COVID-19 patients in Malaysia, and a complete predictive model of anxiety among them. The results showed that the prevalence of anxiety among COVID-19 patients was low. We found that being single/widowed is a significant risk factor for anxiety. The fear of COVID-19, the lack of information regarding COVID-19, behavioral disengagement, and instrumental support were significantly associated with the predictors of anxiety. Additionally, both adaptive and maladaptive coping strategies were associated with anxiety. Findings from this study have the potential to help to improve the current mental health surveillance system and the implementation of appropriate interventions in managing related situations. Finally, this study serves as a springboard for future research on the psychological impact of COVID-19 in Malaysia.

## Figures and Tables

**Table 1 ijerph-20-00586-t001:** Socio-demographic characteristics of respondents (*n* = 401).

Scheme		Number of Respondents (*n*)	Percentage (%)
**Gender**	Male	274	68.3
	Female	127	31.7
**Age group (years)**	18–34	258	64.3
	35–49	106	26.4
	≥50	37	9.2
**Ethnicity**	Malay	274	68.3
	Chinese	27	6.7
	Indian	20	5.0
	Other Bumiputera	7	1.7
	Others	73	18.2
**Citizenship**	Malaysian	328	81.8
	Non-Malaysian	73	18.2
**Marital status**	Married	204	49.1
	Single/widow/er	197	50.9
**Education level**	No formal education	33	8.2
	Primary education	64	16.0
	Secondary education	102	25.4
	Tertiary education	202	50.4
**Occupation**	Civil servant	34	8.5
	Private sector employee	138	34.4
	Self-employed	50	12.5
	Healthcare worker	19	4.7
	Not working/pensioner/student	151	37.7
	Housewife	9	2.2
**Household income group**	B40	288	71.8
	M40	68	17.0
	T20	45	11.2

**Table 2 ijerph-20-00586-t002:** Prevalence of anxiety by socio-demographic characteristics (*n* = 401).

Socio-Demographic Characteristics		Number of Respondents (*n*)	Prevalence (%)	Chi-Square
**Overall**		401	7.0	
**Gender**	Male	20	7.3	0.715
	Female	8	6.3	
**Age group (years)**	18–34	20	7.8	0.718
	35–49	6	5.7	
	≥50	2	5.4	
**Ethnicity**	Malay	23	8.4%	0.231
	Chinese	3	11.1%	
	Indian	0	0.0%	
	Other Bumiputera	0	0.0%	
	Others	2	2.7%	
**Citizenship**	Malaysian	28	7.9%	0.465
	Non-Malaysian	0	0.0%	
**Marital status**	Married	8	4.1%	0.024 *
	Single/widow/er	20	9.8%	
**Education level**	No formal education	1	3.0%	0.447
	Primary education	3	4.7%	
	Secondary education	6	5.9%	
	Tertiary education	18	8.9%	
**Occupation**	Civil servant	1	2.9%	0.054
	Private sector employee	7	5.1%	
	Self-employed	1	2.0%	
	Healthcare worker	0	0.0%	
	Not working/pensioner/student	18	11.9%	
	Housewife	1	11.1%	
**Household income group**	B40	15	5.2%	0.069
	M40	7	10.3%	
	T20	6	13.3%	

* A *p*-value < 0.05 is statistically significant.

**Table 3 ijerph-20-00586-t003:** Simple and multiple logistic regression of anxiety among hospitalized patients with COVID-19.

Predictors	Crude OR (95% CI)	*p*-Value	Adjusted OR (95% CI)	*p*-Value
**Gender**				
Male	1		-	-
Female	0.85 (0.37, 1.99)	0.715	-	-
**Marital Status**				
Single/widowed	2.57 (1.10, 5.98)	0.029	2.87 (1.01, 8.18)	<0.05 *
Married	1		1	
**Education level**				
No formal education	1		-	-
Primary	1.57 (0.16, 15.75)	0.700	-	-
Secondary	2.00 (0.23, 17.25)	0.528	-	-
Tertiary	3.13 (0.40, 24.28)	0.275	-	-
**Occupation**				
Civil servant	0.57 (0.07, 4.77)	0.602	-	-
Private sector employee	1	0.374	-	-
Self-employment	0.38 (0.05, 3.19)	0.374	-	-
Not working/student/pensioner	2.53 (1.02, 6.27)	0.044	-	-
Housewife	2.34 (0.26, 21.40)	0.452	-	-
**Age group (years)**				
18–34	1		-	-
35–49	0.71 (0.28, 1.83)	0.483	-	-
≥50	0.68 (0.15, 3.04)	0.613	-	-
**Household income group**				
B40	0.36 (0.13, 0.98)	0.045	-	-
M40	0.75 (0.23, 2.38)	0.621	-	-
T20	1		-	-
**Stressors**				
Fear of infection	2.20 (1.42, 3.40)	<0.001	1.82 (1.08, 3.04)	<0.05 *
Social discrimination	1.31 (0.90, 1.90)	0.154	-	-
Financial burden	1.31 (0.92, 1.87)	0.138	-	-
Lack of information	1.51 (1.02, 2.22)	0.039	1.68 (1.01, 2.79)	<0.05 *
**Coping strategies**				
Self-distraction	1.38 (1.12, 1.69)	0.002	-	-
Active coping	1.28 (1.04, 1.57)	0.018	-	-
Denial	1.35 (1.06, 1.71)	0.015	-	-
Substance use	2.35 (1.37, 4.01)	0.002	-	-
Emotional support	1.15 (0.95, 1.41)	0.161	-	-
Instrumental support	0.99 (0.81, 1.20)	0.877	0.65 (0.47, 0.90)	<0.001 *
Behavioral disengagement	2.35 (1.74, 3.17)	<0.001	2.03 (1.30, 3.18)	<0.001 *
Venting	1.50 (1.21, 1.87)	<0.001	-	-
Positive reframing	1.12 (0.93, 1.37)	0.240	-	-
Planning	1.15 (0.94, 1.41)	0.185	-	-
Humor	1.78 (1.35, 2.34)	<0.001	-	-
Acceptance	1.11 (0.91, 1.36)	0.321	-	-
Religion	1.02 (0.81, 1.28)	0.863	-	-
Self-blame	2.16 (1.72, 2.72)	<0.001	1.75 (1.27, 2.40)	<0.001 *

Not working/student/pensioner, B40, social discrimination, financial burden, self-distraction, active coping, denial, substance use, emotional support, venting, positive reframing, planning, and humor were removed by SPSS using backward logistic regression analysis. * A *p*-value < 0.05 is statistically significant.

## Data Availability

The data used for this study are not publicly available due to reasons of data protection but are available from the Institute for Public Health, Ministry of Health Malaysia upon reasonable request and with permission from the Director General of Health Malaysia.

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
