# Peer review of "Prevalence and Predictors of Anxiety among Stable Hospitalized COVID-19 Patients in Malaysia"

_ijerph, 2022, doi:10.3390/ijerph20010586_

Round 1

Reviewer 1 Report

The topic of this paper regarding the prevalence and predictors of anxiety among stable hospitalized covid-19 patients in Malaysia is interesting. The paper is generally well-written and seems to share complex information about the topic with respect to the current literature. I have only a minor remark before recommending the manuscript for publication: I do not understand the number 7.3 in Tables 1 and 2 - line citizenship/Non-Malaysian (How can you calculate 7.3 parts of the respondent?).

Author Response

 hi. thank you very much for the feedback.. would be very happy to respond to any inquiry from the reviewer

I do not understand the number 7.3 in Tables 1 and 2 - line citizenship/Non-Malaysian (How can you calculate 7.3 parts of the respondent?).

I think its wrongly written. The actual number of respondent is actual 73 not 7.3. sorry

Reviewer 2 Report

This study aimed to determine the prevalence of anxiety and its associated factors among stable inpatient COVID-19 patients in Malaysia.

Overall, this is an interesting study and quite a well-written manuscript that has the potential to shed more light on the relationship between prevalence of anxiety and pandemic of COVID -19. The aims of the study are not very original. The group sample size is correct. Generally, the introduction and discussion are suitable and literature was referenced in the appropriate context.

My general comment is that this study has potential to add value to the scientific field of health care, however some points need to be re-considered.

Questionnaire
Authors should show the reliability of the Malay version of questionnaire used.

Discussion

Authors should add information about prevalence of anxiety during of Covid 19 pandemic in different part of the world (not only in Asia) and explain the differences during each wave of COVID-19 pandemic. At first time, symptoms of anxiety or depression were at their most severe and probably represented an acute reaction to an unexpected and unknown emerging crisis.

Authors should add new literature f. ex.

The prevalence of depression, anxiety, and sleep disturbances in COVID-19 patients: a meta-analysis. Deng J, Zhou F, Hou W, Silver Z, Wong CY, Chang O, Huang E, Zuo QK. Ann N Y Acad Sci. 2021 Feb;1486(1):90-111. doi: 10.1111/nyas.14506.

Depression and anxiety during COVID-19. Daly M, Robinson E Lancet. 2022 Feb 5;399(10324):518. doi: 10.1016/S0140-6736(22)00187-8.

Depressive and anxiety symptoms and COVID-19-related factors among men and women in Nigeria. Oginni OA, Oloniniyi IO, Ibigbami O, Ugo V, Amiola A, Ogunbajo A, Esan O, Adelola A, Daropale O, Ebuka M, Mapayi B.

PLoS One. 2021 Aug 26;16(8):e0256690. doi: 10.1371/journal.pone.0256690. eCollection 2021.

Author Response

Hi

first all of , thank you very much for reviewing my article. ill try my very best to answer your review. thanks

  1. Questionnaire
    Authors should show the reliability of the Malay version of questionnaire used.
  • Both GAD 7 and Brief Cope (malay version) has been shown to have good validity and reliability. ill add it to the article during my revision.

    - https://pubmed.ncbi.nlm.nih.gov/22377544/
    - https://journals.lww.com/mjp/Abstract/2009/18010/RELIABILITY_AND_VALIDITY_OF_THE_MALAY_VERSION_OF.5.aspx

    2.Discussion
    Authors should add information about prevalence of anxiety during of Covid 19 pandemic in different part of the world (not only in Asia) and explain the differences during each wave of COVID-19 pandemic. At first time, symptoms of anxiety or depression were at their most severe and probably represented an acute reaction to an unexpected and unknown emerging crisis

  • thanks for the suggestion. ill look up some articles regarding the matter and add them to my article. thanks.

Reviewer 3 Report

Dear Authors,

Thank you for the opportunity to review your manuscript focusing on prevalence and predictors of anxiety among stable hospitalized COVID-19 patients in Malaysia. Assessing mental health and coping strategies are important especially in patient with COVID-19, since this pandemic has put tremendous pressure on individuals as well the society at large due to socioeconomic disruption, fear and associated uncertainties thereby impacting the mental health. This trend appears to be continuing as we are still not completely out of the woods.  Current study attempted to assess the prevalence and predictors of anxiety  among hospitalized patient with COVID infection in Malaysia. There are number of methodological issues that certainly deserves authors’ attention to improve their current manuscript. Appended below are the major and minor comments on the manuscript.     

The current manuscript requires extensive English language editing, use of more appropriate and widely accepted terms, editing for sentence composition and text flow, formatting etc. Authors are advised to seek help appropriate help.   

1.         Since, the instruments were administered online, how the issues such as duplicate responses and mailbox stuffing were avoided.

2.         How authors came up with a sample size of 401 to be appropriate for this study?

3.         Although, authors stated that the GAD-7 was validated previously, no information about its reliability or validity was provided in the manuscript. The study referenced for GAD-7 validation in Malay, also did not provide any data on its reliability or validity measures. In the absence of such measures, it is unlikely if this instrument is even valid to be used, especially the Malay version. The reference provided is also wrong, since Sherina et al. listed under the references is nowhere to be found.

4.         Similarly, the Malay version of the Brief-Cope (Coping Orientation to Problems Experienced) Inventory was initially validated among adolescents from Secondary School. A totally different setting that the current population where it was used. Moreover, according to the authors of this validation study of its Malay version, three coping strategies items viz. venting of emotion, denial and acceptance demonstrated poor reliability and suggested that these items must be revised in order to improve this instrument’s reliability.   

5.         Considering the above issues with both instruments and settings of its current use, in addition to the lack of results from pilot testing of these instrument in current setting, the result of this study appears to be questionable.

6.         The GAD-7 score of 2.58 with a SD of 3.58 clearly suggest large variability in responses, further indicating potential floor and ceiling effect. This may have very well confounded  the estimation of anxiety prevalence.  

7.          Readers could have been benefited more and had better understanding by having granular data about both instruments with their binary scores and distribution across the socio-demographic characteristics in Table 2.

8.         Authors should use at least two decimal points or present prevalence as a rate in Table 2 to provide the actual prevalence rather than simply reporting “0”, which in fact is not absolutely a “zero” prevalence.  

9.         Authors should also elaborate on household income presented as B40, M40 and T20, and how it corresponds to actual currency/monetary value or equivalent dollar amount for the readers of this international journal.

10.   Please replace term “correlated” while desorbing the results of aOR, as it indicates association and not “correlation”.

11.    Data presented in Table 3 appears to be confusing. It is not clear in this table which variables were retained for multivariate regression analysis after univariate. It would be better if the results are presented as model 1, model 2 and so on.  

12.    Authors should also provide some greater details about their regression analyses results including the Chi-square statistic for overall model fit, pseudo R2,  etc.

13.    Considerable number of references are drawn from web pages, some are even older than 10-15 years.      

Author Response

  1. Since, the instruments were administered online, how the issues such as duplicate responses and mailbox stuffing were avoided

- For this matter, 2 of our research assistant has been assigned to the designated hospital to distrube the google survey form to the qualified respondents. In the form, respondent background such bed number, ward and name of hospital were taken. This is to minimize the duplicate responds issue. While our central team will download the the responses in the google server daily

Refer here - https://iku.gov.my/images/IKU/Document/REPORT/covid-19/Covid-19_Technical_Reports_part1-1.pdf

  1. How authors came up with a sample size of 401 to be appropriate for this study?

- The sample size was calculated using the Sample Size Calculation Formula for a prevalence with finite population correction study as per the primary objective.The calculation is done with a margin of error of 0.05 and Type 1 error determined at 5% with a finite population of 500.The largest sample size obtained was 214 respondents. Multiplying non-response and those who refuse to participate by 30%, the final sample size was determined to be 278 respondents per hospital

Reference - https://iku.gov.my/images/IKU/Document/REPORT/covid-19/Covid-19_Technical_Reports_part1-1.pdf

  1. Although, authors stated that the GAD-7 was validated previously, no information about its reliability or validity was provided in the manuscript. The study referenced for GAD-7 validation in Malay, also did not provide any data on its reliability or validity measures. In the absence of such measures, it is unlikely if this instrument is even valid to be used, especially the Malay version. The reference provided is also wrong, since Sherina et al. listed under the references is nowhere to be found.

Both GAD 7 and Brief Cope (malay version) has been shown to have good validity and reliability. ill add it to the article during my revision.

- https://pubmed.ncbi.nlm.nih.gov/22377544/

- https://journals.lww.com/mjp/Abstract/2009/18010/RELIABILITY_AND_VALIDITY_OF_THE_MALAY_VERSION_OF.5.aspx

Ill make changes to manuscript and include the necessary information

  1. Similarly, the Malay version of the Brief-Cope (Coping Orientation to Problems Experienced) Inventory was initially validated among adolescents from Secondary School. A totally different setting that the current population where it was used. Moreover, according to the authors of this validation study of its Malay version, three coping strategies items viz. venting of emotion, denial and acceptance demonstrated poor reliability and suggested that these items must be revised in order to improve this instrument’s reliability.  

- This paper validates the Malay Version of Brief COPE Scale. The result shown that Brief COPE Scale (Malay Version) confirms fairly good reliability and validity.

- https://journals.lww.com/mjp/Abstract/2009/18010/RELIABILITY_AND_VALIDITY_OF_THE_MALAY_VERSION_OF.5.aspx

  1. Considering the above issues with both instruments and settings of its current use, in addition to the lack of results from pilot testing of these instrument in current setting, the result of this study appears to be questionable.

    - Based from the technical report , at section 3.8 , it was stated that , Pilot study with minimum 30 different responses was conducted to validate the system (google form) and revision has been made as per the feedback.

    Refer here - https://iku.gov.my/images/IKU/Document/REPORT/covid-19/Covid-19_Technical_Reports_part1-1.pdf
  2. The GAD-7 score of 2.58 with a SD of 3.58 clearly suggest large variability in responses, further indicating potential floor and ceiling effect. This may have very well confounded  the estimation of anxiety prevalence.

Noted. I will go through the data analysis again

  1. Readers could have been benefited more and had better understanding by having granular data about both instruments with their binary scores and distribution across the socio-demographic characteristics in Table 2.

    - After discussion with my team members, we did not include the data for both instruments as we are focusing on the prevalence and logistic regression of the study.

  1. Authors should use at least two decimal points or present prevalence as a rate in Table 2 to provide the actual prevalence rather than simply reporting “0”, which in fact is not absolutely a “zero” prevalence.  

- Noted. Ill will amend it in my revision.

  1. Authors should also elaborate on household income presented as B40, M40 and T20, and how it corresponds to actual currency/monetary value or equivalent dollar amount for the readers of this international journal.

- Noted. Ill make sure to elaborate this point in my revision

  1. Please replace term “correlated” while desorbing the results of aOR, as it indicates association and not “correlation”.

-  Noted. Il makes changes to this mistake.

  1. Data presented in Table 3 appears to be confusing. It is not clear in this table which variables were retained for multivariate regression analysis after univariate. It would be better if the results are presented as model 1, model 2 and so on.  

All variables with P-value < 0.25 in univariable analysis and variables known to be associated with anxiety from published articles were included in the model to control possible confounding factors. The backward LR step was used to get the best predictor model for anxiety. We cant show all step as they are like 19 steps to reach the final step.

  1. Authors should also provide some greater details about their regression analyses results including the Chi-square statistic for overall model fit, pseudo R2, etc.

We asses the goodness of fit by

  1. The hosmer-Lemeshow test. The p-value in Hosmer is >0.05, which is 0.673, assumption is met. The model is fit.
  2. The overall correctly classified percentage is 95%, which is >70%. Assumption is met. Model is fit
  3. Area under the Receiver Operating Characteristic (ROC) curve is 0.931 (95% CI:0.88,0.98). It is significantly different from 0.5 (p-value <0.05). The model can accurately discriminate 93.1% of the cases. which is >70%. Assumptions are met. Final model is achieved
  4. Considerable number of references are drawn from web pages, some are even older than 10-15 years.      

Web pages citation is only included if it deems necessary. And it’s a reliable source of information for citation. Such as from WHO, Malaysia covid cases and technical report. Additionally, old references are cited to relate to the current topic which is about endemic/diseases

Thank you very much for the feedback. Im sorry for the late reply as I was busy

Round 2

Reviewer 3 Report

Dear Authors, 

Thank you for the revision. 

Author Response

My manuscript has undergone proofreading and English language editing by a local lecturer who has a certificate for it. Thank you also for reviewing my paper, and you are most welcome.